# Plant-Derived Antimicrobials and Their Crucial Role in Combating Antimicrobial Resistance

**DOI:** 10.3390/antibiotics13080746

**Published:** 2024-08-09

**Authors:** Paola Angelini

**Affiliations:** Department of Chemistry, Biology and Biotechnology, University of Perugia, 06122 Perugia, Italy; paola.angelini@unipg.it

**Keywords:** antimicrobial resistance, plant metabolites, plant-derived antibiotics, artificial intelligence

## Abstract

Antibiotic resistance emerged shortly after the discovery of the first antibiotic and has remained a critical public health issue ever since. Managing antibiotic resistance in clinical settings continues to be challenging, particularly with the rise of superbugs, or bacteria resistant to multiple antibiotics, known as multidrug-resistant (MDR) bacteria. This rapid development of resistance has compelled researchers to continuously seek new antimicrobial agents to curb resistance, despite a shrinking pipeline of new drugs. Recently, the focus of antimicrobial discovery has shifted to plants, fungi, lichens, endophytes, and various marine sources, such as seaweeds, corals, and other microorganisms, due to their promising properties. For this review, an extensive search was conducted across multiple scientific databases, including PubMed, Elsevier, ResearchGate, Scopus, and Google Scholar, encompassing publications from 1929 to 2024. This review provides a concise overview of the mechanisms employed by bacteria to develop antibiotic resistance, followed by an in-depth exploration of plant secondary metabolites as a potential solution to MDR pathogens. In recent years, the interest in plant-based medicines has surged, driven by their advantageous properties. However, additional research is essential to fully understand the mechanisms of action and verify the safety of antimicrobial phytochemicals. Future prospects for enhancing the use of plant secondary metabolites in combating antibiotic-resistant pathogens will also be discussed.

## 1. Introduction

In the early 20th century, pharmacologist Alexander Fleming serendipitously discovered penicillin, the first natural antibiotic compound [1]. Interestingly, antibiotic resistance was identified in the same year, 1928 [2]. This groundbreaking discovery revolutionized the medical field, paving the way for the development of other antibiotics such as streptomycin, chloramphenicol, erythromycin, and chlortetracycline. Between 1960 and 1980, the pharmaceutical industry experienced a “golden period” in antibiotic development, producing numerous new antimicrobials [3].

After this period, the development of new antibiotics significantly declined as the pharmaceutical industry shifted its focus towards more profitable drug categories. This shift, combined with the rise in antimicrobial resistance, restricted the development of new antimicrobial agents [4,5]. Consequently, the increasing incidence of bacterial infections, coupled with their evolving multidrug resistance, has transformed them into major global health threats. The Comprehensive Antibiotic Resistance Database (CARD) reveals the existence of 5159 reference sequences. Although 381 of these are linked to relatively few significant pathogens, this offers little relief [6]. The situation is further exacerbated by the slow rate at which new generations of therapeutically effective antibiotics become available [7].

Antimicrobial resistance (AMR) refers to the ability of microorganisms, including bacteria, viruses, fungi, and parasites, to survive and multiply despite drug treatment [8]. Antimicrobial agents, such a antibiotics, antivirals, and antifungals, are designed to inhibit or eliminate pathogens. Among these, antibiotics are the most commonly used, particularly for bacterial infections. AMR develops naturally through genetic mutations, allowing organisms to adapt and survive [8]. In response to environmental pressures, bacteria evolve mechanisms to resist antimicrobial drugs, diminishing the effectiveness of these treatments [9]. The widespread use of antibiotics, especially in low-resource settings, provides bacteria with numerous opportunities to develop resistance, leading to significant health consequences, such as increased morbidity and mortality [10,11,12,13].

The rapid global rise of “superbugs” (microorganisms resistant to most available antimicrobials) underscores the urgent issue of drug-resistant pathogens. The World Health Organization (WHO) has identified AMR as one of the top three critical threats to global public health [14]. A report from 2019 estimated that antimicrobial-resistant infections were directly responsible for approximately 1.27 million deaths, and a study published in January 2022 indicated that nearly 5 million additional deaths were associated with drug-resistant infections [15]. Projections suggest that by 2050, the annual death toll from such infections could rise to 10 million, far surpassing cancer fatalities [13].

Methicillin-resistant *Staphylococcus aureus* (MRSA) exemplifies an early “superbug”, significantly contributing to global mortality rates from drug-resistant infections [16]. Bacteria, present in both domestic and professional environments, are among the earliest life forms on Earth [17]. While most bacterial species are harmless to human health, specific strains like *Staphylococcus aureus*, *Helicobacter pylori*, *Escherichia coli*, and *Bacillus anthracis* can breach host defenses and cause severe diseases such as pneumonia, endocarditis, septicemia, and osteomyelitis [18].

Healthcare-associated infections continue to pose a significant threat to patient safety and public health, often leading to severe complications and imposing considerable societal burdens [19]. Traditional preventive strategies for clinical infections primarily rely on aseptic techniques and systemic antibiotic therapies, but these methods frequently prove inadequate [20]. For instance, systemic antibiotic therapy for infections linked to medical devices, such as catheters, artificial prosthetics, subcutaneous sensors, and orthopedic implants, has an effectiveness rate of only 22–37% [21]. Moreover, high doses of antibiotics required for localized infections can result in cytotoxicity and adverse effects on surrounding tissues [22], further accelerating the emergence of bacterial drug resistance [23,24].

To combat infections caused by drug-resistant bacteria, scientists have developed various antimicrobial agents, including antibacterial peptides, amphiphiles, and antimicrobial materials such as nanoparticles, hydrogels, engineered surfaces, and coatings [25,26]. Despite these advancements, bacterial resistance remains a significant challenge. Current research is focused on discovering methods to eliminate bacteria without promoting resistance.

## 2. Methodology

To conduct this review, a comprehensive search was performed across several scientific databases, including PubMed, Elsevier, ResearchGate, Scopus, and Google Scholar. The search covered publications from 1929 to 2024, ensuring both a broad historical perspective and contemporary insights. A total of 278 publications were collected and thoroughly assessed.

The search strategy was designed to encompass a wide range of relevant literature, utilizing English MeSH (Medical Subject Headings) descriptors to ensure consistency and comprehensiveness. To maintain the quality and relevance of the selected studies, several criteria were applied. Only articles published in English that focused on the antibacterial properties of medicinal plant extracts were included. Duplicate articles, review articles, and studies evaluating isolated, commercially acquired molecules were excluded. This ensured that the review concentrated solely on the effects of plant extracts.

By implementing these methods, this review aimed to provide a thorough and detailed analysis of the antibacterial properties of medicinal plants, particularly their efficacy against species identified as priorities by the World Health Organization.

## 3. Strategies for Overcoming Antibiotic Resistance

The misuse and overuse of antibiotics has led to the emergence of multidrug-resistant bacteria, or superbugs, complicating infection treatment in clinical settings. This escalating issue necessitates innovative treatment strategies, which in turn require a comprehensive understanding of the mechanisms pathogenic bacteria employ to resist antibiotics.

### 3.1. Antibiotic Modification

Antibiotics can be categorized based on their targets: the bacterial cell envelope, the cytoplasm during protein synthesis, or the bacterial genome. To counteract antibiotics, bacteria produce modifying enzymes. For instance, bacteria resistant to β-lactam antibiotics produce β-lactamase, which hydrolyzes the β-lactam ring, rendering antibiotics like penicillins, carbapenems, monobactams, and cephalosporins ineffective.

Based on primary sequence homology, β-lactamase enzymes are grouped into four classes: A, B, C, and D. While classes A, C, and D are serine-based, class B is metal-based and includes a Zn^2+^ ion. Serine-based β-lactamases form a covalent acyl-enzyme intermediate with the antibiotic, which is then hydrolyzed. In contrast, class B β-lactamases use water molecules for hydrolysis without forming an intermediate [27].

To combat β-lactamase activity, one strategy involves combining β-lactam antibiotics with β-lactamase inhibitors like clavulanic acid and tazobactam. These inhibitors bind to the enzyme with higher affinity, allowing the antibiotic to remain effective. Another approach is modifying the antibiotic structure, such as removing the aminoadipoyl sidechain from cephalosporins to form 7-aminocephalosporanic acid, making them significantly more effective and resistant to β-lactamase hydrolysis.

### 3.2. Modification of Antibiotic Target Sites

Microorganisms employ various strategies to withstand antibiotic treatment, with one of the most effective being the modification of the antibiotic’s target site. This approach involves a minor gene mutation that alters the enzyme’s target site, reducing the binding affinity of antibiotics while maintaining the enzyme’s functionality. For example, β-lactam antibiotics like penicillin target and inactivate penicillin-binding protein (PBP), a transpeptidase that assists in cross-linking the peptidoglycan cell wall by acylating its active site. A mutation in the PBP gene can result in a slightly altered but still functional protein, preventing β-lactam antibiotics from binding effectively. Drug-resistant strains of *Clostridium difficile*, *Enterococcus faecium*, and *Streptococcus pneumoniae* exhibit this ability [28,29]. Microbial resistance to ciprofloxacin, which interferes with cellular division by targeting DNA gyrase and topoisomerase, also falls under this mechanism [30,31].

Another method of conferring resistance is through post-transcriptional and post-translational modifications of antibiotic target sites. Post-transcriptional modifications involve altering the primary RNA transcript to form mature RNA, which can then be translated into functional proteins. For instance, methylation of the tRNA anticodon stem loop prevents frame-shifting during translation. Methylation of specific 16S ribosomal RNA has been reported to prevent the binding of antibiotics like spectinomycin and streptomycin, thus allowing normal protein translation [32,33]. Post-translational modifications involve enzymatic alterations of proteins after synthesis. Common modifications in bacteria include phosphorylation and succinylation. For example, cysteine phosphorylation in eukaryotic-like kinase-phosphatases has been shown to confer resistance to vancomycin and ceftriaxone in methicillin-resistant *Staphylococcus aureus* (MRSA) [34]. Additionally, lysine succinylation in isocitrate lyase has been linked to resistance to rifampicin in *Mycobacterium tuberculosis*.

### 3.3. Antibiotic Resistence: Efflux Pump and Reduced Permeability

Bacteria have developed various strategies to counteract antibiotics by preventing these drugs from accumulating sufficiently within their cytoplasm. These strategies include overproducing efflux pumps in the bacterial cytoplasmic membrane and reducing membrane permeability.

Efflux pumps are active transport proteins that expel toxic compounds, including antibiotics, from the bacterial cytoplasm. They use energy derived from adenosine triphosphate (ATP) or an electrochemical potential gradient. For instance, bacteria such as *Escherichia coli* and *Pseudomonas aeruginosa* show significant resistance to ciprofloxacin and fluoroquinolones due to the overexpression of efflux pump proteins powered by the hydrogen ion gradient [35,36]. Instead of having pumps specific to certain antibiotics, bacteria like MRSA and *P. aeruginosa* produce non-specific multidrug resistance pumps that expel a wide range of antibiotics, including β-lactams [37,38,39].

Another resistance mechanism involves reducing the permeability of the bacterial outer membrane. Bacteria transport small polar molecules, amino acids, and nutrients through water-filled channels called porins [40]. These channels also permit the entry of antibiotics like penicillin and cephalosporins. To counteract this, bacteria can either decrease or silence the expression of porin proteins or alter their structure to reduce permeability. For example, drug-resistant strains of *E. coli* and *Enterobacter aerogenes* modify the structure of porins by narrowing the channels, thereby limiting antibiotic entry. Conversely, *Klebsiella pneumoniae* reduces antibiotic uptake by minimizing porin expression [41,42,43].

### 3.4. Enhancing Resistance through Target/Substrate Overproduction

The discovery of antibiotic modifiers like the β-lactamase enzyme paved the way for combinatorial treatments that pair antibiotics with β-lactamase inhibitors. However, it is not surprising that bacteria eventually become resistant to these inhibitors, especially with the frequent use of such combinations. This resistance arises in addition to the high mutation rates bacteria experience due to the direct effects of both the antibiotic and the inhibitor. Consequently, bacteria can develop resistance to β-lactamase inhibitors through the overproduction of the target enzyme, β-lactamase. This phenomenon has been observed in drug-resistant strains of *E. coli* and *K. pneumoniae* [44,45].

Another example is the overexpression of the enzyme dihydrofolate reductase, which is essential for nucleic acid precursor synthesis in *M. tuberculosis* and *E. coli*. Overproduction of this enzyme confers resistance to antimicrobial drugs such as para-aminosalicylic acid and trimethoprim [46,47].

### 3.5. Cell Wall Remodeling

Under typical physiological conditions, bacterial cell walls undergo constant remodeleling, with old peptidoglycan layers being degraded and replaced by new ones. Glycopeptide antibiotics primarily disrupt this cell wall construction process. For instance, teicoplanin and vancomycin inhibit the actions of glycosyltransferases and transpeptidases by binding to the peptidyl-D-Ala-D-Ala ends of peptidoglycan precursors [48,49]. This binding blocks transpeptidation and transglycosylation, preventing the bacterial cell from renewing its peptidoglycan wall. As the old wall breaks down due to treatment, the cell ultimately undergoes lysis and dies.

However, vancomycin-resistant *Enterococci* species can produce altered peptidoglycan precursors, such as peptidyl-D-Ala-D-Lac or peptidyl-D-Ala-D-Ser, which impede the binding of glycopeptide antibiotics [50,51]. These modifications arise through either gene acquisition or inherent genetic mutations. Specifically, the vanA gene cluster encodes VanH dehydrogenase, which converts pyruvate into D-Lac, and VanA ligase, which esterifies D-Ala with D-Lac to form peptidyl-D-Ala-D-Lac [52]. Such changes prevent the binding of glycopeptide antibiotics, while allowing glycosyltransferases and transpeptidases to continue synthesizing the cell wall.

Moreover, a single amino acid alteration in the Dd1 ligase, responsible for ligating two D-Ala molecules, can enable it to connect D-Ala to D-Lac instead, thereby conferring resistance to glycopeptide antibiotics. This mechanism has also been observed in vancomycin-resistant strains of *E. coli*.

## 4. Plant-Derived Antibiotics: A Solution to Multidrug-Resistant Microbes

Exploring innovative antibiotics derived from natural products is crucial for addressing the socio-economic and health impacts of multidrug-resistant microbes [53]. Plant-derived therapeutic agents have gained importance due to the emergence of new diseases and the growing scientific understanding of herbal medicines as alternative or complementary treatments [54] (Figure 1). Several plants and plant parts known for their antimicrobial properties and available on the market are enumerated in Table 1.

Research shows that medicinal plants contain bioactive compounds such as coumarins, flavonoids, phenolics, alkaloids, terpenoids, tannins, essential oils, lectins, polypeptides, and polyacetylenes [56,57,58], which serve as foundations for antibiotic development [58]. For instance, crude extracts from *Polygonum persicaria*, *P. plebejum*, *Rumex hastatus*, *R. dentatus*, *R. nepalensis*, and *Rheum australe* exhibit antibacterial and antifungal properties, inhibiting bacteria such as *C. frundii*, *E. coli*, *E. aerogenes*, and *S. aureus* [59]. Additionally, *Calotropis gigantea* extracts have shown significant antifungal activity against pathogenic fungi like *Candida albicans* and *Aspergillus* species in Asia [57]. Ethanolic extracts from *Plumbago zeylanica* roots exhibit strong antimicrobial effects against *V. cholerae*, *E. coli*, *P. aeruginosa*, *Curvularia lunata, Colletotrichum corchori*, and *Fusarium equiseti* [58]. The aqueous leaf extracts of *Euphorbia hirta* and *Erythrophleum suaveolens*, along with the methanolic leaf extract of *Thevetia peruviana*, show antibacterial effects against extended-spectrum beta-lactamase (ESBL)-producing bacteria, including *E. coli*, *Pseudomonas*, *K. pneumonia*, methicillin-resistant *Staphylococcus aureus* (MRSA), *Salmonella*, and *Proteus* [60,61,62,63]. Limited research on aqueous and hydro-alcoholic extracts from various plants has revealed antibacterial effects on multidrug-resistant bacteria, including MRSA and ESBL producers [64].

Methanolic and ethyl acetate extracts from *Anacyclus maroccanus* Ball and *A. radiatus* Loisel have been evaluated for their antimicrobial activity against a variety of bacterial, fungal, and dermatophyte species. Notably, *E. coli* and *T. rubrum* exhibited the highest sensitivity to these extracts [65]. The antimicrobial characteristics of aqueous extracts derived from the roots and leaves of *Lactuca longidentata* have been analyzed in relation to various microorganisms, including Gram-positive and Gram-negative bacteria, yeast strains from the *Candida* genus, dermatophytes such as species from the *Trichophyton* and *Arthroderma* genera, and fungi commonly found as contaminants in public swimming pools. Within the limits of biocompatibility (concentration < 200 µg/mL), the leaf extract demonstrated notable efficacy in inhibiting the growth of *Escherichia coli* and *Trichophyton tonsurans*, with a minimum inhibitory concentration (MIC) of less than 10 µg/mL [66]. The root and leaf extracts, both aqueous and hydroalcoholic, from female *Cannabis sativa* cv. *Strawberry* exhibit potent antimicrobial activity against *Bacillus subtilis* (PeruMycA 6). Most tested bacterial strains were highly susceptible to both hydroalcoholic extracts, which demonstrated MIC values below 62.99 µg/mL. These extracts also showed significant efficacy in inhibiting dermatophyte growth, with *Arthroderma currey* (CCF 5207) being the most sensitive fungal species, exhibiting MIC values less than 6.25 µg/mL [67]. These findings align with previous studies on the antifungal effects of water extracts from the inflorescences of the industrial hemp (*Cannabis sativa* L.) cultivar “Futura 75” [68].

Despite the approval of synthetic antimicrobial agents in numerous countries, natural compounds derived from plants [69,70], fungi [71], lichens [72], endophytes [73], and various marine sources, such as plants [74,75,76,77], seaweeds [78], corals [79], and other microorganisms [80,81], remain a focal point of substantial research interest [70]. These natural compounds have shown promise in combating antibiotic resistance in bacterial pathogens [82]. Among the various options, plant-derived compounds are notable for their potential in fighting bacterial infections. These naturally occurring plant chemicals have demonstrated significant benefits, including antioxidant, antibacterial, and antifungal activities. They can enhance the effectiveness of existing antibiotics, helping to prevent the development of resistance [83].

Based on their chemical structures, these compounds can be categorized into major groups such as alkaloids, sulfur-containing compounds, terpenoids, and polyphenols.

**Table 1 antibiotics-13-00746-t001:** Some of the plant products with antimicrobial activity. This table is adapted from Khameneh et al. [84].

Family	Scientific Name (Common Name)	Compound	Effective in Combating	Drug Delivery System
Berberidaceae	*Berberis vulgaris* (Barberry)	Berberine	Bacteria, protozoa	Soft gel 1000 mg
Piperaceae	*Piper nigrum* (Black pepper)	Piperine	Fungi, *Lactobacillus*, *Micrococcus*	
Asteraceae	*Arctium lappa* (Burdock)		Bacteria, fungi, virus	Capsule 475 mg
Apiaceae	*Carum carvi* (Caraway)		Bacteria, fungi, virus	Capsule 1000 mg
Rhamnaceae	*Rhamnus purshiana* (Cascara sagrada)	Tannins	Bacteria, fungi, virus	Capsule 425, 450 mg
Asteraceae	*Matricaria chamomilla* (Chamomille)	Anthemic acid	*M. tuberulosis*, *S. typhimurium*, *S. aureus*	
Apiaceae	*Syzygium aromaticum* (Clove)	Eugenol	General	Capsule 500 mg
Ericaceae	*Vaccinium* spp. (Cranberry)	Fructose	Bacteria	Capsule 500 mg
Myrtaceae	*Eucalyptus globulus* (Eucalyptus)	Tannins	Bacteria, virus	Inhaler and tablet
Amaryllidaceae	*Allium sativum* (Garlic)	Allicin, ajoene	General	Tablet
Asteraceae	*Hydrastis canadensis* (Goldenseal)	Berberine, hydrastine	Bacteria, *Giarda duodenale*, *Trypanosomes*	Solution, 500 mg per dosage
Theaceae	*Camellia sinensis* (Green tea)	Catechin	General	
Fabaceae	*Glycyrrhiza glabra* (Licorice)	Glabrol	*S. aureus*, *M. tuberculosis*	Capsule 450 mg
Fagaceae	*Quercus rubra* (Oak)	Tannins, Quercetin		Capsule 500, 650 mg
Amaryllidaceae	*Allium cepa* (Onion)	Allcin	Bacteria, *Candida*	
Berberidaceae	*Mahonia aquifolia* (Oregon grape)	Berberine	*Plasmodium*, *Trypansomes*, general	Capsule 500 mg
Hypericaceae	*Hypericum perforatum* (Senna St. John’s wort)	Hypericin, others	General	Capsule 450 mg
Lamiaceae	*Thymus vulgaris* (Thyme)	Caffeic acid, Thymol, Tannins	Viruses, bacteria, fungi	Capsule 450 mg
Zingiberaceae	*Curcuma longa* (Turmeric)	Curcumin, Turmeric oil	Bacteria, protozoa	

## 5. Plant Secondary Metabolites as Antimicrobial Agent

### 5.1. Alkaloids

The term “alkaloid” denotes “similar to alkalis”, indicating the basic or alkaline nature of these substances. To date, around 12,000 alkaloid compounds have been extracted from plants and categorized. These compounds exhibit diverse medicinal properties, including antitumor, analgesic (such as morphine and codeine), and antimicrobial effects [85]. Their chemical structures feature heterocyclic rings with N-heterocyclic nitrogen and can be classified based on their carbon precursors and structural characteristics [86]. Their antibacterial activity is well-documented, and many studies suggest they play a crucial role in treating infectious diseases [87]. Most alkaloids function as efflux pump inhibitors (EPIs), which represents a key antibacterial mechanism [88].

Piperine, a piperidine-type alkaloid from *Piper nigrum* and *Piper longum*, when combined with ciprofloxacin, has been shown to inhibit the growth of a mutant *S. aureus* and significantly reduce its MIC values. Additionally, the co-administration of piperine and gentamicin proved effective against MRSA infections. Studies have shown that piperine impacts the NorA efflux pump activity in both *S. aureus* and MRSA [88,89].

Berberine, an isoquinoline alkaloid, is found in the roots and stem-bark of *Berberis* species. It is the primary active ingredient in *Rhizoma coptidis* and *Cortex phellodendri* and has long been utilized in traditional medicine. This compound has demonstrated effectiveness against a variety of pathogens, including bacteria, fungi, protozoa, and viruses. It intercalates DNA, targets RNA polymerase, gyrase, and topoisomerase IV, and inhibits cell division. Berberine also inhibits the FtsZ protein, essential for bacterial cell division, and disrupts cell structure [90,91,92,93].

Ungeremine, an isoquinoline alkaloid extracted from *Pancratium illyricum* L. bulbs, possesses significant antibacterial properties. It enhances DNA cleavage by targeting and inhibiting bacterial topoisomerase IA [94,95].

Quinoline alkaloids like dictamnine, koku-sagine, and maculine, extracted from the stem bark of *Teclea afzeli*, have shown significant antibacterial properties. These compounds inhibit type II topoisomerase enzymes, DNA replication, and act as respiratory inhibitors [96,97].

Reserpine, an indole alkaloid obtained from *Rauwolfia serpentina*, is a well-known natural compound with strong EPI activity. When combined with reserpine, various bacterial species, including *Staphylococcus*, *Streptococcus*, and *Micrococcus*, have demonstrated increased antibiotic susceptibility [98,99]. It enhances antibiotic susceptibility in various bacteria by inhibiting efflux pumps and is effective against MDR *Acinetobacter baumannii* and *Stenotrophomonas maltophilia* [100,101,102].

Sanguinarine, extracted from specific plants such as *Chelidonium majus*, *Sanguinaria canadensis*, and *Macleaya cordata*, disrupts bacterial membranes, acts as a DNA intercalator, and inhibits replication and transcription. It also shows antimycobacterial activity [103,104,105].

Tomatidine, a steroidal alkaloid found in solanaceous plants like tomatoes, potatoes, and eggplants, boosts the effectiveness of antibiotics against *S. aureus* and other bacteria, particularly when used with aminoglycosides [106].

Chanoclavine, a tricyclic ergot alkaloid derived from *Ipomoea muricata*, acts synergistically with tetracycline to combat MDR *E. coli* by inhibiting ATPase-dependent efflux pumps [107].

Conessine, a steroidal alkaloid derived from *Holarrhena antidysenterica* barks, is effective against both Gram-positive and Gram-negative bacteria, showing synergistic effects with conventional antibiotics. It acts as an EPI against *A. baumannii* [108,109,110].

Squalamine, a natural steroid-polyamine compound first isolated from the dogfish shark, is distinct from many other compounds, as it is not primarily plant-derived. It disrupts microbial membranes by interacting with their components, leading to cell death in both Gram-negative and Gram-positive bacteria. These alkaloids not only boost the effectiveness of conventional antibiotics, but also offer potential solutions to combat antibiotic resistance [111].

### 5.2. Organosulfur Compounds

A significant body of literature discusses the antibacterial and antifungal properties of plant-derived sulfur-containing compounds [82,112,113]. Various compounds, including allicin, ajoene, dialkenyl and dialkyl sulfides, S-allyl cysteine, S-allyl-mercapto cysteine, and isothiocyanates, have demonstrated significant antibacterial properties against a wide range of bacteria, both Gram-positive and Gram-negative [114,115]. Research indicates that plants with high polysulfide content exhibit broad-spectrum antimicrobial properties [116,117].

Allicin (allyl 2-propenethiosulfinate or diallyl thiosulfinate or S-allyl cysteine sulfoxide), derived from garlic (*Allium sativum*) and other *Allium* species, has been found to have antimicrobial effects against bacteria such as *Staphylococcus epidermidis*, *Pseudomonas aeruginosa*, *Streptococcus agalactiae*, MRSA, and oral pathogens responsible for periodontitis [118]. Allicin has been shown to boost the effectiveness of antibiotics like cefoperazone, tobramycin, and ciprofloxacin against *P. aeruginosa* [119]. Its antimicrobial properties arise from the inhibition of sulfhydryl-dependent enzymes, such as alcohol dehydrogenase, thioredoxin reductase, and RNA polymerase [120]. The presence of cysteine and glutathione diminishes allicin’s inhibitory impact, suggesting that these compounds interact with allicin to mitigate microbial damage. Additionally, allicin partially inhibits DNA and protein synthesis and may also affect RNA [120].

Ajoene, another compound from garlic, is composed of the E- and Z-ajoene stereoisomers. It demonstrates broad-spectrum antimicrobial activity, effective against bacteria, fungi, and protozoa, and possesses even stronger antiviral properties compared to allicin [121]. Like allicin, the inhibitory effect of ajoene is diminished in the presence of cysteine, suggesting that both compounds share a mechanism that targets thiol-dependent enzyme systems [122].

Isothiocyanates (ITCs) are volatile sulfur compounds produced from plant glucosinolates and the enzyme myrosinase, found in *Brassicaceae* plants like cauliflower, cabbage, mustard, and broccoli. ITCs have potent antibacterial effects and are considered promising candidates. For instance, ITCs derived from horseradish (*Armoracia rusticana*) roots demonstrate potent antimicrobial effects against oral pathogens [123,124]. They act as highly effective bactericides against *Helicobacter pylori*, functioning by inhibiting urease and reducing inflammation [125]. The antimicrobial action of ITCs is likely attributed to their interaction with proteins, thereby disrupting essential biochemical processes. Their primary mode of action involves targeting sulfhydryl groups by reacting with amines, thiols, and hydroxyls. Additionally, ITCs inhibit bacterial ATP binding sites by targeting cysteine residues [125].

Sulforaphane, present in various plants like *Diplotaxis harra*, is derived from 4-methyl sulfinyl butyl glucosinolate. It exhibits strong anticarcinogenic and antibacterial properties, particularly against *H. pylori*, a known risk factor for stomach cancer. Additionally, sulforaphane is effective against *S. aureus* and *L. monocytogenes*, highlighting its potential as a natural antibacterial agent [126,127].

Allyl isothiocyanates (AITCs), found in *Brassicaceae* plants such as *Armoracia rusticana* and *Eutrema japonicum*, possess significant antibacterial properties against *E. coli* and *S. aureus*, demonstrating both bacteriostatic and bactericidal activities [128]. AITCs lower the minimum inhibitory concentration (MIC) values of erythromycin against *S. pyogenes* and show synergistic effects with streptomycin against *E. coli* and *P. aeruginosa* [129,130]. However, AITCs exhibit low inhibitory effects against certain Gram-positive bacteria [131,132]. Their antimicrobial mechanisms include compromising cell wall integrity and inducing internal structural changes, as observed through electron microscopy [133]. AITCs can additionally deactivate crucial intracellular enzymes by oxidatively cleaving disulfide bonds [134]. They form pores in bacterial cell membranes, causing leakage of intracellular substances [135].

Benzyl isothiocyanate (BITC), found in *Alliaria petiolate* [136], has demonstrated effectiveness against 15 MRSA isolates, exhibiting bactericidal effects on 11 of them. This suggests BITC’s potential in combating MRSA strains [137]. The antibacterial action of BITC is attributed to its lipophilic and electrophilic properties, which enable it to penetrate bacterial membranes and disrupt their integrity [138].

Phenethyl isothiocyanate (PEITC), present in *Brassica* vegetables such as *Brassica campestris* and *Brassica rapa* [136], exhibits antimicrobial activity primarily against Gram-positive bacteria, with lower efficacy against Gram-negative ones [137,138]. Additionally, PEITC has antifungal properties against *Alternaria brassicicola* [139], possibly due to mechanisms involving reduced oxygen consumption, intracellular ROS accumulation, and mitochondrial membrane depolarization [140].

Berteroin, a compound present in broccoli (*Brassica oleracea* L.), exhibits the lowest minimum inhibitory concentration (MIC) values against both extracellular and intracellular bacterial strains, highlighting it as a highly potent bactericidal compound. It also proves effective against *H. pylori* [141,142].

### 5.3. Phenolic Compounds

Phenolic compounds encompass a wide array of bioactive natural molecules extensively employed in medical applications. These compounds enhance the effectiveness of antibiotics against resistant pathogens by employing multiple mechanisms [143,144,145]. One notable mechanisms is the reduction of efflux pump (EP) activity, acting as efflux pump inhibitors (EPIs) and demonstrating significant EPI activity against pathogenic bacteria. Table 1 lists important plant-derived EPIs.

Resveratrol, a natural phenolic compound, inhibits the CmeABC efflux pumps of *Campylobacter jejuni* and the efflux pumps of *Mycobacterium smegmatis* [146,147]. Ferreira et al. [147] reported that resveratrol increases ethidium bromide accumulation in *Arcobacter butzleri* and *Arcobacter cryaerophilus* [148].

Baicalein, a flavone from *Thymus vulgaris*, *Scutellaria baicalensis*, and *Scutellaria lateriflora*, enhances the effectiveness of β-lactam antibiotics, tetracycline, and ciprofloxacin against MRSA by inhibiting the NorA efflux pump [149]. Combining baicalein with tetracycline exhibits synergistic effects against *E. coli* by inhibiting efflux pumps [150,151].

Biochanin A, an isoflavone, inhibits the efflux system of MRSA by reducing NorA protein expression and shows potent activity against *Chlamydia* spp. and *Mycobacterium* strains [152,153,154]. Other flavonoids, such as chrysosplenol-D and chrysoplenetin from *Artemisia annua*, inhibit the NorA efflux pump in the presence of subinhibitory berberine concentrations [155]. Isoflavonoids and flavonolignans also inhibit NorA, enhancing the potency of norfloxacin and berberine [156].

Silybin, biochanin A, genistein, and orobol augment the activity of *S. aureus* against NorA substrates [156,157]. The hybridization of antibiotics with flavonoids diminishes efflux pump activity, thereby boosting antibiotic accumulation and efficacy.

Kaempferol has emerged as a promising candidate against various pathogens, effective against fluconazole-resistant *C. albicans* and MRSA by inhibiting the NorA pump [158,159]. Kaempferol rhamnoside from *Persea lingue* increases ciprofloxacin’s antimicrobial activity in NorA-overexpressing *S. aureus* strains [160]. Brown et al. [159] identified efflux pump inhibitors using an LC-MS method, finding that rhamentin and kaempferol significantly inhibit efflux pumps [161].

Quercetin exhibits moderate efflux pump inhibition, more effectively identified with LC-MS methods than fluorescence-based techniques [159].

Chalcones, such as 4′,6′-Dihydroxy-3′,5′-dimethyl-2′-methoxychalcone from *Dalea versicolor*, inhibit the NorA efflux pump and reduce erythromycin MIC [159]. Holler et al. [161] identified two synthetic chalcones with potent NorA efflux pump inhibitory activity.

Catechin gallates, like epigallocatechin gallate (EGCG), provide health benefits and potent antimicrobial activity against resistant pathogens, including MRSA, by weakly inhibiting the NorA efflux pump [162]. Several phenolic compounds inhibit DNA gyrase, such as novobiocin [163,164], and natural products like green tea polyphenols, chebulinic acid, and anthraquinones exhibit similar activity [165,166,167]. EGCG inhibits the B subunit of DNA gyrase and efflux pumps, making it a significant focus for future research.

Chebulinic acid from *Terminalia chebula* effectively inhibits quinolone-resistant *M. tuberculosis* DNA gyrase [163]. Haloemodins, which are semisynthetic anthraquinone derivatives, inhibit DNA gyrase in MRSA and vancomycin-resistant *Enterococcus faecium* [163].

3-p-trans-Coumaroyl-2-hydroxyquinic acid (CHQA) from *Cedrus deodara* demonstrates potent antibacterial activity against food-borne pathogens by damaging the cytoplasmic membrane and inducing intracellular leakage [168].

Hydroxycinnamic acids like p-coumaric acid interfere with membrane integrity, displaying significant activity [169].

Flavonoids engage with critical enzymes involved in the production of bacterial cell membrane precursors, including beta-ketoacyl acyl carrier protein synthase (KAS) II and III. Flavanones like naringenin, eriodictyol, and taxifolin inhibit KAS III, exhibiting moderate antibacterial activity [170]. 3,6-Dihydroxyflavone binds to KAS I and III, indicating that many flavonoids can inhibit these enzymes. EGCG covalently binds to and inactivates beta-ketoacyl[acyl carrier protein] reductase (FabG) [171].

Sakuranetin exhibits potent antibacterial activity by inhibiting FabZ in *Helicobacter pylori*. Quercetin and apigenin also inhibit FabZ, though less effectively [172]. Curcumin damages the cell membranes of *S. aureus* and *E. coli* due to its amphipathic and lipophilic structure [173].

Quercetin and apigenin inhibit d-alanine ligase in *H. pylori* and *E. coli*, albeit with high MIC values, indicating low inhibitory activity [174].

Sophoraflavanone B directly interacts with peptidoglycan, inhibiting MRSA growth [175]. Other phenolic compounds inhibit enzymes like dihydrofolate reductase, urease, and sortase [176,177,178].

Curcumin inhibits sortase A in *S. aureus* [179,180]. Morin from *Rhus verniciflua* inhibits sortase A and B, showing fibrinogen cell-clumping activity [180,181,182]. Flavonoids like 4′,7,8-trihydroxy-2-isoflavene inhibit urease, offering potential as natural inhibitors [168].

Phenolic compounds exhibit diverse mechanisms against bacteria, from efflux pump inhibition and cell membrane interaction to enzyme inhibition. Their significant activities make them promising candidates for future studies and clinical trials. EGCG and curcumin, with their multiple modes of action, exemplify compounds that bacteria cannot easily resist.

### 5.4. Coumarins

Coumarins are naturally produced by many plants and microorganisms [181,182]. They possess a wide range of bioactivities, including vasodilator, estrogenic, anticoagulant, analgesic, anti-inflammatory, sedative, hypothermic, anti-helminthic, anticancer, antioxidant, and dermal photosensitizing properties [183,184]. Numerous studies have highlighted the antimicrobial activity of both natural and synthetic coumarin derivatives [185,186]. For example, Basile et al. [187] identified several coumarins and pyranocoumarins from *Ferulago campestris* roots, such as agasyllin, grandivittin, and aegelinol benzoate, which showed antibacterial and antioxidant activities against both Gram-negative and Gram-positive bacteria.

Aegelinol and agasyllin showed significant effectiveness against *Salmonella enterica* serovar *typhi*, *Enterobacter aerogenes*, *E. cloacae*, and *S. aureus*, with minimum inhibitory concentrations (MIC) of 16 μg/mL for aegelinol and 32 μg/mL for agasyllin. Additionally, they exhibited dose-dependent activity against *Helicobacter pylori* at concentrations ranging from 5 to 25 μg/mL [185,188,189].

Tan et al. [186] identified one new and nine previously known prenylated coumarins in extracts from the roots of *Prangos hulusii* and assessed their antimicrobial efficacy against both standard strains and clinical isolates. The new coumarin, 4′-senecioiloxyosthol, was highly active against *Bacillus subtilis* (MIC = 5 μg/mL), while osthole was effective against *B. subtilis*, *S. aureus*, *Klebsiella pneumonia*, and methicillin-sensitive *Staphylococcus aureus* (MSSA) (all MICs = 125 μg/mL).

El-Seedi [190] identified a new aryl coumarin glucoside, asphodelin A 4′-O-β-D-glucoside, and its aglycon, asphodelin A, from *Asphodelus microcarpus*. Asphodelin A exhibited potent activity against *S. aureus*, *E. coli*, *P. aeruginosa*, *C. albicans*, and *Botrytis cinerea*, with MIC values ranging from 4 to 128 μg/mL.

Maxwell [191] investigated the structure-activity relationship (SAR) of coumarins such as clorobiocin, novobiocin, and coumermycin A1, which are derived from different *Streptomyces* species. His findings indicated that the noviosyl sugar moiety, in conjunction with the coumarin structure, is essential for biological activity. These coumarins are highly effective inhibitors of DNA topoisomerase type II, commonly referred to as DNA gyrase [192].

Structure-activity relationship (SAR) studies have demonstrated that lipophilicity and a planar configuration are crucial for potent antibacterial activity [193]. The antimicrobial efficacy of coumarins is primarily due to passive diffusion, which enhances cellular penetration, especially in Gram-positive bacteria. Sardari et al. [194] suggested that the presence of a free 6-OH group is important for antifungal activity, while a free 7-OH group is vital for antibacterial activity. Further systematic SAR analysis revealed that coumarins possessing a methoxy group at C-7 and a hydroxyl group at C-6 or C-8 exhibit broad-spectrum antibacterial properties. An aromatic dimethoxy configuration boosts effectiveness against certain pathogens, such as *Haemophilus influenzae*, beta-hemolytic *Streptococcus*, and *Streptococcus pneumoniae* [193]. Recent studies also suggest that coumarins can suppress bacterial quorum-sensing networks, affecting biofilm formation and virulence factor production [183,195,196,197,198,199].

Some coumarin derivatives can inhibit EP in MRSA strains. Bergamottin epoxide, a furanocoumarin from grapefruit (*Citrus paradisi*), reduced the MIC value of norfloxacin against MRSA by 20-fold via EP inhibition [198]. Another study on EP inhibition by coumarins from *Mesua ferrea* found two compounds that reduced the MIC of norfloxacin by 8-fold in MRSA and clinical *S. aureus* isolates [200].

Coumarins can bind to isoprene units in plant cells to form more complex structures. 6-Geranyl coumarin and galbanic acid are terpenoid coumarins that significantly inhibit EP in *S. aureus*. Galbanic acid reduced the MIC of ciprofloxacin by up to 8-fold, with a mode of action comparable to verapamil, a known EP inhibitor [201].

### 5.5. Terpenes

Terpenes, also known as isoprenoids, constitute the most diverse group of natural compounds and are present in almost every form of life. They serve various functions, ranging from contributing to cell structure (such as cholesterol and steroids in membranes) to facilitating cellular processes (such as retinal in vision, carotenoids in photosynthesis, and quinones in electron transport) [202,203].

Terpenes are plentiful in flowers, fruits, and vegetables, particularly in plant reproductive structures and foliage during and after the flowering phase. They constitute a significant part of herbal resins and are responsible for the distinctive scents of many plants [203]. Numerous terpenes and their derivatives function as protective agents against herbivores and pathogens [204]. Gram-positive bacteria are generally more susceptible to terpenes than Gram-negative ones due to terpenes’ lipophilic properties. Monoterpenes increase membrane fluidity and permeability, alter protein topology, and disrupt the respiration chain [205].

Togashi et al. [206] studied the inhibitory effects of various terpene alcohols on *S. aureus*, finding that farnesol and nerolidol had strong antibacterial effects with minimum bactericidal concentration (MBC) values of 20 and 40 μg/mL, respectively. They further investigated how these terpene alcohols interact with bacterial cell membranes by evaluating intracellular K+ ion leakage, indicating that potassium leakage serves as an indicator of the antibacterial strength of compounds that disrupt cell membranes. Farnesol and nerolidol were the most effective, with the hydrocarbon chain length between C10 and C12 being optimal for antibacterial and membrane-disrupting activity [207]. Dehydroabietic acid, a resin acid, also shows antibacterial activity against *S. aureus* [208,209,210,211].

Carvone, particularly (4R)-(−)-carvone, is effective against *Campylobacter jejuni*, *E. faecium*, and *E. coli*, while (4S)-(+)-carvone is active against *L. monocytogenes*. Both isomers demonstrate efficacy against a range of pathogenic fungi, inhibiting the transformation of *C. albicans* from yeast to its pathogenic filamentous form [212,213,214].

Thymol shows strong efficacy against *Candida albicans*, *C. glabrata*, and *C. krusei*, both independently and when combined with fluconazole. MIC values are 49.37, 51.25, and 70 μg/mL, respectively. Thymol shows synergistic effects with fluconazole against *Candida* species and, along with carvacrol, eugenol, and menthol, is effective against various food-decaying fungi, making them good alternatives to synthetic fungicides [215].

Thymol and carvacrol, major components of *Thymus capitatus*, also exhibit antibacterial effects against *E. coli*, *Enterobacter aerogenes*, *S. aureus*, and *P. aeruginosa*, with MIC values of 0.005–0.008 mg/mL for thymol and 0.007–0.008 mg/mL for carvacrol [216,217]. These compounds reduce bacterial counts on polypropylene surfaces during the biofilm formation of *Salmonella* spp. (*S. typhimurium*, *S. enteritidis*, and *S. saintpaul*) and disrupt established biofilms at MIC or 2 × MIC [218]. Thymol’s mechanism against *S. typhimurium* involves “disruption of membrane integrity”, while thymol and carvacrol inhibit EP in a concentration-dependent manner, enhancing ethidium bromide accumulation in foodborne pathogens [219].

Ursolic acid and α-amyrin, two pentacyclic triterpenes, show broad-spectrum antibacterial activity by disorganizing the *E. coli* inner membrane model [220]. Eugenol and cinnamaldehyde, found in essential oils, prevent *H. pylori* growth and exhibit bioactivity against MRSA and MSSA biofilms, disrupting cell-to-cell communication and biofilm construction [221]. Cinnamaldehyde damages bacterial membrane structure, reduces membrane potential, and affects metabolic activity, inhibiting *E. coli* and *S. aureus* growth [222,223,224]. Various terpenoid derivatives exhibit antimycobacterial properties against pathogens like *M. tuberculosis*.

Copp [222] documented that a range of terpenoids, such as sandaracopimaric acid, (+)-totarol, agelasine F, elisapterosin B, costunolide, parthenolide, 1,10-epoxycostunolide, santamarine, reynosin, alantolactone, puupehenone, elatol, deschloroelatol, debromolaurinterol, allolaurinterol, and aureol, are effective against *M. tuberculosis*. This activity is linked to their lipophilic structures, aiding in the penetration of the mycobacterial cell wall.

### 5.6. Antimicrobial Peptides from Plants

Plants are abundant sources of biologically active compounds with a wide range of properties, many of which find applications in medicine and agriculture [224,225,226,227]. Among these, antimicrobial peptides (AMPs) stand out. AMPs are small, amphiphilic molecules that generally weigh between 2 and 10 kDa, and typically carry a positive charge at neutral and physiological pH levels [228,229]. Despite significant variations in their primary and secondary structures, most plant-derived AMPs share a compact spatial structure stabilized by intramolecular disulfide bonds, which provide stability against temperature fluctuations, enzymatic activity, and chemical agents [228,230].

Plant AMPs are grouped into various families based on their amino acid sequences, cysteine motifs, disulfide bond locations, and secondary structural elements [228,229]. The main AMP families include defensins, thionins, α-hairpinins (hairpin-like peptides), hevein-like peptides, knottins, snakins, lipid-transfer proteins, and cyclotides. There are also peptides that do not fit into these categories, such as those with atypical cysteine motifs, those without disulfide bonds, cyclic peptides lacking a cysteine knot, and peptides that are rich in glycine, histidine, and alanine [228,230,231,232,233,234,235]. According to the Data Repository of Antimicrobial Peptides (DRAMP), more than 800 plant peptides have been identified so far [236].

Antimicrobial peptides (AMPs) are essential components of plant innate immunity, particularly in defense against biotic stressors [230]. These peptides demonstrate a variety of functions, such as antibacterial, antifungal, insecticidal, and antiviral activities. Additionally, certain AMPs are known to inhibit hydrolases and protein synthesis [237]. Owing to their chemical characteristics, plant AMPs also possess antiproliferative properties [238,239]. These properties make AMPs promising candidates for the development of novel pharmaceuticals or biological agents for plant protection [240].

AMPs are present in all plants, and each species within a specific taxon (such as a genus or family) displays a distinct molecular diversity of peptides from various structural families [241,242,243,244]. They can be extracted from various parts of plants, including vegetative [245] and generative parts [242,246,247,248,249,250], as well as from aboveground [242,248,249] and underground tissues [250,251,252,253]. Seeds, in particular, are a rich source of diverse AMPs [234].

Advancements in transcriptomic and proteomic methods have significantly enhanced plant AMP research [254,255,256]. However, the isolation of AMPs remains crucial for studying their structure–function relationships at the cellular and organism levels [257,258,259]. Classical extraction methods continue to be relevant, especially for isolating new peptides that are not yet classified within existing plant AMP families [229,232,251].

Summarizing the accumulated knowledge and developing comprehensive isolation schemes that allow for the extraction of a wide variety of peptides will provide new opportunities in AMP research.

## 6. The Utility of Artificial Intelligence (AI)

Numerous pathogenic bacteria have increasingly developed resistance to existing antibiotics, while the pace of new antibiotic development has significantly slowed. This escalating resistance poses a severe threat to global health, underscoring the urgent need for innovative solutions. 

Leveraging AI algorithms offers a novel approach to accelerating the drug discovery process. In a recent study, researchers trained a neural network to screen approximately 7500 molecules, resulting in the identification of a compound named abaucin. This compound proved effective in controlling an *Acinetobacter baumannii* infection [260]. This success highlights AI’s potential to identify promising antibiotic candidates more efficiently than traditional methods.

AI, especially through machine learning and deep learning techniques, is instrumental not only in designing new antibiotics but also in optimizing the effectiveness of existing drugs through synergistic combinations [261]. Machine learning algorithms analyze patterns in antimicrobial resistance (AMR) data, providing critical support to healthcare providers and policymakers by predicting which bacteria and fungi may develop resistance to specific drugs or compounds [185,262]. Furthermore, machine learning models are valuable tools for AMR surveillance. By examining data on antimicrobial usage and the presence of resistant microorganisms, these models allow public health authorities to make well-informed decisions, prepare for, and respond swiftly to resistance outbreaks by identifying emerging resistance patterns and at-risk populations and areas [11]. Collectively, these applications play a crucial role in mitigating the overall impact of AMR [263].

## 7. Other Strategies to Overcome Antibiotic Resistance

Researchers are actively exploring natural resources to find viable alternatives to traditional antibiotics. Plants, in particular, are considered a valuable source of antimicrobial compounds [264,265]. Despite identifying numerous phytochemicals, many remain unexplored. The primary challenge lies not only in discovering these compounds but also in translating laboratory findings into practical applications within clinical and hospital settings. Bridging the gap between research and real-world healthcare implementation of these natural antimicrobial agents is a significant obstacle that scientists are striving to overcome [266,267].

One promising strategy focuses on increasing the intracellular concentration of antimicrobials within bacterial cells. This can be achieved through potentiation by targeting non-essential bacterial components, such as efflux pump inhibitors [268].

Another approach involves using membrane transporters, like iron transporters, by conjugating antibiotics to an iron-binding siderophore mimetic group, facilitating their entry and enhancing their biological activity [269].

A recent study identified di-berberine conjugates that demonstrate enhanced synergistic effects with aminoglycosides, highlighting their potential as lead compounds in the development of efflux pump inhibitors (EPIs) [270].

## 8. Future Perspectives and Conclusions

The increasing evidence supporting the efficacy of medicinal plants in treating infectious diseases underscores their potential as sources of novel antimicrobial agents [271]. These natural compounds, especially phytochemicals, exhibit significant antimicrobial activity against various human pathogens. Their benefits include accessibility, cost-effectiveness, and minimal side effects. Additionally, some plant-derived compounds can enhance the effectiveness of traditional antibiotics, offering a promising strategy to combact bacterial resistance [272].

A key approach involves the co-administration of plant compounds with conventional antibiotics, which can reduce minimum inhibitory concentration (MIC) values and produce synergistic effects. This method parallels successful antibiotic combinations like amoxicillin-clavulanate and isoniazid-rifampicin-pyrazinamide-ethambutol, which target different bacterial sites to enhance efficacy and prevent resistance development [273].

Understanding the specific molecular mechanisms of antimicrobial plants is crucial for developing new therapeutic strategies [274]. Proposed mechanisms include disrupting bacterial cell membranes, inhibiting efflux pumps (EPs), and blocking DNA and protein synthesis [275]. For instance, the combination of EGCG and tetracycline has shown synergistic effects by inhibiting bacterial efflux pumps and restoring antibiotic efficacy in resistant strains [276].

The rapid evolution of resistance among bacteria and other microbes poses a significant threat to modern medicine, challenging the effectiveness of existing antimicrobial therapies [277]. The overuse of antibiotics in healthcare and agriculture has exacerbated this issue, leading to the proliferation of multidrug-resistant pathogens. As antibiotic discovery lags behind the pace of resistance development, we risk entering a post-antibiotic era where common infections could become untreatable [278].

Addressing antimicrobial resistance (AMR) requires comprehensive and coordinated global action. This includes implementing stewardship programs to limit inappropriate antibiotic use, enhancing infection control measures, and fostering international cooperation [279]. A One Health approach, integrating human, animal, and environmental health, is essential for curbing resistance transmission and preserving the efficacy of antimicrobials [280].

Future research should focus on identifying and developing more potential treatments from plant secondary metabolites. Combining these natural compounds with existing antibiotics holds promise for effectively treating infections caused by resistant pathogens. With continued innovation and global collaboration, we can mitigate the impact of AMR and safeguard public health.

## Figures and Tables

**Figure 1 antibiotics-13-00746-f001:**
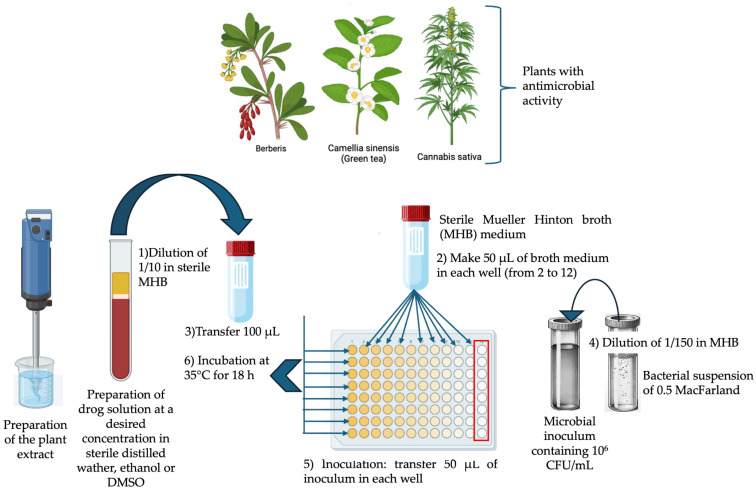
Broth microdilution for antibacterial testing as recommended by the Clinical and Laboratory Standards Institute (CLSI) M07-A9: Methods for Dilution Antimicrobial Susceptibility Tests for Bacteria That Grow Aerobically; Approved Standard—Ninth Edition [55].

## Data Availability

No new data were created or analyzed in this study.

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
