# Peer review of "Plant-Derived Antimicrobials and Their Crucial Role in Combating Antimicrobial Resistance"

_antibiotics, 2024, doi:10.3390/antibiotics13080746_

Round 1

Reviewer 1 Report

Comments and Suggestions for Authors

This review concentrates solely on the antimicrobial effect of plant extracts. Along with describing future prospects for enhancing the use of secondary plant metabolites in combating antibiotic-resistant pathogens. 

However, some justification and clarification need to be addressed.

1. In methodology section, authors have mentioned that "The search covered publications from 1929 to 2024, ensuring a broad historical perspective and contemporary insights." However, later it is stated that "This targeted search was conducted within the PubMed databases and was limited to articles published between 2018 and 2024, ensuring the inclusion of the most recent and relevant findings". Please justify this discrepancy in search year.

2. The search covered publications from 1929 to 2024 only a total of 278 publications were collected. it is not justified.

Author Response

Comment 1: 

This review concentrates solely on the antimicrobial effect of plant extracts. Along with describing future prospects for enhancing the use of secondary plant metabolites in combating antibiotic-resistant pathogens. However, some justification and clarification need to be addressed.

In methodology section, authors have mentioned that "The search covered publications from 1929 to 2024, ensuring a broad historical perspective and contemporary insights." However, later it is stated that "This targeted search was conducted within the PubMed databases and was limited to articles published between 2018 and 2024, ensuring the inclusion of the most recent and relevant findings". Please justify this discrepancy in search year.

The search covered publications from 1929 to 2024 only a total of 278 publications were collected. it is not justified. There are no prospects for the future. The drawings are typically copied and it looks bad. I do not accept the article for publication.

Response to Reviewer: Thank you for pointing this out. I agree with this comment. Therefore, I have eliminated the following sentence: “This targeted search was conducted within the PubMed databases and was restricted to articles published between 2018 and 2024, ensuring the inclusion of the most recent and relevant findings.”– page number 2, paragraph “2. Methodology”]

Reviewer 2 Report

Comments and Suggestions for Authors

line 76 coma at the end of the line

line  92-93 is double of the line 73-74

line 103-104 is double of the line 109-110

line 134 typo error

line 175, 179, 187 tab

line 276 explain Futura 75

line 686-688 689-691 source missing, probably also in the list of references

There is no real conclusion. Section after line 693 should be extended.

Author Response

Comments:

line 76 coma at the end of the line

Response to Reviewer: Agree, it has been corrected.

line 92-93 is double of the line 73-74

Response to Reviewer: Agree. I have, accordingly, eliminated lines 92-93.

line 103-104 is double of the line 109-110

Response to Reviewer: Agree. I have, accordingly, eliminated lines 109-110

line 134 typo error

Response to Reviewer: Agree. I have, accordingly, corrected the error.

line 175, 179, 187 tab

Response to Reviewer: Agree. I have, accordingly, corrected the tab.

line 276 explain Futura 75

Response to Reviewer: Agree. I have, accordingly, corrected as follows: industrial hemp, Cannabis sativa L. cultivar ‘Futura 75’.

line 686-688 689-691 source missing, probably also in the list of references

Response to Reviewer: Thank you very much for the valuable advice. References 271 and 272 have been added in lines 686-688, 689-692, and in the list of references.

271_Rodríguez, D.; González-Bello, C. Siderophores: Chemical tools for precise antibiotic delivery. BMCL 2023, 87, 129282.

272_Morita, Y.; Nakashima, K-I; Nishino, K.; Kotani, K.; Tomida, J.; Inoue, M.; Kawamura, Y. Berberine Is a Novel Type Efflux Inhibitor Which Attenuates the MexXY-Mediated Aminoglycoside Resistance in Pseudomonas aeruginosa. Front. Microbiol. 2016, 7, 1223.

There is no real conclusion. Section after line 693 should be extended.

Response to Reviewer:  would like to express my gratitude to the Reviewer for the valuable comments, which have significantly enhanced the quality of the manuscript. Accordingly, the section “8. Future Perspectives and Conclusions” has been added.

Reviewer 3 Report

Comments and Suggestions for Authors

The subjuect of the manuscript is important. It would be very important to find or exploit subtitutes of antibiotics or new combinations of substance with antibiotics. The author must comment on oregano and carvacrol and other plants of labiatae family. The author has not used significant references on the combined use of antiobiotics and phenolic compounds both in vivo and vitro and their tests as antimicrobials and the reduction or not of the resistance to antibiotics by phenolic comounds.

Comments on the Quality of English Language

The use of English is acceptable. However, the similarity rate of the manuscript is high more than 40%. The author must reduce it by less than 15% in the text (excluding references)

Author Response

Comments:

The subject of the manuscript is important. It would be very important to find or exploit subtitutes of antibiotics or new combinations of substance with antibiotics. The author must comment on oregano and carvacrol and other plants of labiatae family. The author has not used significant references on the combined use of antibiotics and phenolic compounds both in vivo and vitro and their tests as antimicrobials and the reduction or not of the resistance to antibiotics by phenolic compounds.

Response to Reviewer: Agree. I have, accordingly, replaced references 24, 63, 68, 78, and 85 with references concerning the combined use of antibiotics and phenolic compounds both in vivo and in vitro.

The use of English is acceptable. However, the similarity rate of the manuscript is high more than 40%. The author must reduce it by less than 15% in the text (excluding references).

Response to Reviewer: Agree. Accordingly, I have made several modifications to the manuscript, which are highlighted in the track changes of the re-submitted files.

Reviewer 4 Report

Comments and Suggestions for Authors

In the review paper, the author briefly discussed the basic mechanisms used by bacteria to develop resistance to antibiotics. In the next part of the manuscript, however, she presented the characteristics of phytochemicals that could find potential use in combating microbes resistant to multiple antibiotics. Among plant secondary metabolites, she discussed alkaloids, coumarins, terpenes, peptides as well as organosulfur and phenolic compounds.

The issues presented in the paper are very important and meet the need for new substances with potential antimicrobial activity. Pointing to secondary plant metabolites is a step in the right direction due to the fact that secondary metabolites are a very large and chemically diverse group of natural substances with relatively low toxicity to the human body. The information on plant metabolites contained in the paper is presented in an understandable way and based on reliable scientific literature (278 references). It should be noted that the latest cited articles are from 2024.

The weakness of the work is due to the fact that, except for one table (Table 1), the manuscript lacks any other illustrative material. It is quite tedious to read a dozen pages of a publication consisting in practice only of text.

Lines 462 and 465, “LC-MS method” instead of “LC-Mass-based method”.

Author Response

Comments:

In the review paper, the author briefly discussed the basic mechanisms used by bacteria to develop resistance to antibiotics. In the next part of the manuscript, however, she presented the characteristics of phytochemicals that could find potential use in combating microbes resistant to multiple antibiotics. Among plant secondary metabolites, she discussed alkaloids, coumarins, terpenes, peptides as well as organosulfur and phenolic compounds.

The issues presented in the paper are very important and meet the need for new substances with potential antimicrobial activity. Pointing to secondary plant metabolites is a step in the right direction due to the fact that secondary metabolites are a very large and chemically diverse group of natural substances with relatively low toxicity to the human body. The information on plant metabolites contained in the paper is presented in an understandable way and based on reliable scientific literature (278 references). It should be noted that the latest cited articles are from 2024.

The weakness of the work is due to the fact that, except for one table (Table 1), the manuscript lacks any other illustrative material. It is quite tedious to read a dozen pages of a publication consisting in practice only of text.

Response to Reviewer: Agree. Accordingly, I have added Figure 1: Broth microdilution for antibacterial testing as recommended by the Clinical and Laboratory Standards Institute (CLSI) in the M07-A9 protocol (2012).

Lines 462 and 465, “LC-MS method” instead of “LC-Mass-based method”

Response to Reviewer: Agree. Accordingly, "LC-Mass-based method" has been replaced with "LC-MS method".

Round 2

Reviewer 3 Report

Comments and Suggestions for Authors

Authors have revised adequately. But authors should reduce similarity by less than 20%.